# Revisiting the global mean ocean mass budget over 2005-2020

Anne Barnoud[1], Julia Pfeffer[1], Anny Cazenave[1,2], Robin Fraudeau[1], Victor Rousseau[1], and
Michaël Ablain[1]

[1]Magellium, 31520 Ramonville-Saint-Agne, France
[2]LEGOS, Toulouse, France

**Correspondence:** Anne Barnoud (anne.barnoud@magellium.fr)

**Abstract.** We investigate the performances of GRACE and GRACE Follow-On satellite gravimetry missions in assessing the
ocean mass budget at global scale over 2005-2020. For that purpose, we focus on the last years of the record (2015-2020)
when GRACE and GRACE Follow-On faced instrumental problems. We compare the global mean ocean mass estimates from
GRACE and GRACE Follow-On to the sum of its contributions from Greenland, Antarctica, land glaciers, terrestrial water
storage and atmospheric water content estimated with independent observations. Significant residuals are observed in the
global mean ocean mass budget at interannual time scales. Our analyses suggest that the terrestrial water storage variations
based on global hydrological model likely contributes to a large part to the misclosure of the global mean ocean mass budget
at interannual time scales. We also compare the GRACE-based global mean ocean mass with the altimetry-based global mean
sea level corrected for the Argo-based thermosteric contribution (an equivalent of global mean ocean mass). After correcting
for the wet troposphere drift of the radiometer on-board the Jason-3 altimeter satellite, we find that mass budget misclosure
is reduced but still significant. However, replacing the Argo-based thermosteric component by the ORAS5 ocean reanalysis
or from CERES top of the atmosphere observations significantly reduces the residuals of the mass budget over the 2015-2020
time span. We conclude that the two most likely sources of error in the global mean ocean mass budget are the thermosteric
component based on Argo and the terrestrial water storage contribution based on global hydrological models. The GRACE and
GRACE Follow-On data are unlikely to be responsible on their own for the non-closure of the global mean ocean mass budget.

## 1 Introduction

The increase in ocean mass due to land ice melting was responsible for about two thirds of the global mean sea level rise over
2006-2018 (Fox-Kemper et al., 2021), which has major impacts for the populations living in coastal areas (e.g. IPCC, 2019;
Horwath et al., 2022). Since 2002, the Gravity Recovery and Climate Experiment (GRACE; Tapley et al., 2019) and GRACE
Follow-On (GRACE-FO; Landerer et al., 2020) satellite gravimetry missions allow monitoring the ocean mass variations from
space as the gravity field is directly sensitive to the redistribution of water masses on land and in the oceans. These data are
used to assess and understand the effects of climate change and climate variability on the Earth system, such as variations
of freshwater storage (e.g. Vishwakarma et al., 2021), ice sheet melting (e.g. Velicogna et al., 2020; Shepherd et al., 2021),
interannual variability of water mass transport (Pfeffer et al., 2022a), variations of Earth's energy imbalance (Hakuba et al.,
2021; Marti et al., 2022). Ensuring the stability of GRACE and GRACE-FO data is therefore very important for climate

and hydrological applications. Both missions have encountered some instrumental problems due to battery and accelerometer failures (Bandikova et al., 2019). Moreover, the one-year gap between the GRACE and GRACE-FO missions has led to missing data between mid-2017 (end of GRACE life) and mid-2018 (launch of GRACE-FO data). Despite these issues, no bias between the two subsequent missions was reported by comparing GRACE and GRACE-FO data to independent estimates for specific components such as ice sheet mass loss (Velicogna et al., 2020) or terrestrial water storage variations (Landerer et al., 2020).

However, Chen et al. (2020) reported a non-closure of the global mean sea level budget as of 2016 by comparing the global mean ocean mass (GMOM) variations based on GRACE and GRACE-FO data to the altimetry-based global mean sea level (GMSL) variations corrected for the Argo-based global mean steric sea level variations. While 40 % of the non-closure was identified as the result of errors in salinity measurements of the Conductivity-Temperature-Depth sensors (CTDs) of the Argo network (Barnoud et al., 2021), part of the non-closure remained unexplained and potentially due to other components of the sea level budget, including the GRACE and GRACE-FO-based ocean mass. In this study, we investigate whether the GRACE and GRACE-FO mass component could be responsible for the remaining non-closure of the GMSL budget observed over the most recent years (Barnoud et al., 2021). We focus on the recent years (beyond 2015) noting that the sea level and ocean mass budgets were successfully shown to be closed within uncertainties until 2016 (Horwath et al., 2022). Using state-of-the-art datasets, we assess the global mean ocean mass budget from January 2005. We compare the GRACE and GRACE-FO-based GMOM with the sum of individual mass contributions from independent data sources available until December 2018. These mass components include ice-mass loss from the ice sheets, ice caps and glaciers, and terrestrial water storage changes. We also compare the GMOM with the altimetry-based GMSL corrected for thermosteric effects until December 2020 using three different datasets for the latter (Argo data, ORAS5 ocean reanalysis and top of the atmosphere CERES data expressed in terms of thermosteric contribution to sea level).

## 2   Method

### 2.1   Global mean ocean mass budget approach

The ocean mass change, or barystatic sea level change (Gregory et al., 2019), refers to the sea level change due to the freshwater fluxes between the oceans on the one hand and continents and atmosphere on the other hand. The ocean mass budget consists in comparing the ocean mass change with independent observations. The global mean ocean mass change $\Delta GMOM$ can be broken down into its contributions as follows:

$$\Delta GMOM = \Delta GIS + \Delta AIS + \Delta GIC + \Delta TWS + \Delta AWV + \epsilon_1 \tag{1}$$

where $\Delta GIS$, $\Delta AIS$, $\Delta GIC$, $\Delta TWS$ and $\Delta AWV$ refer to Greenland and Antarctica ice sheets mass loss, glaciers and ice caps melting, terrestrial water storage changes and atmospheric water vapour content changes. $\epsilon_1$ accounts for other potentially negligible contributions (e.g. permafrost thawing) and data errors.

GMOM variations can also be estimated from GMSL changes corrected for the global mean steric sea level change due to temperature and salinity variations. At global scale, the mean halosteric sea level change due to salinity variations is negligible

(Gregory and Lowe, 2000; Llovel et al., 2019), so that the global mean steric sea level change is nearly fully accounted for by the global mean thermosteric sea level changes $\Delta GMTSL$. Therefore, the $\Delta GMOM$ can be written as:

$$\Delta GMOM = \Delta GMSL - \Delta GMTSL + \epsilon_2 \tag{2}$$

where $\epsilon_2$ accounts for potentially negligible remaining contributions, including the nearly null global mean halosteric sea level, and errors (e.g. due to the evolution of the deep ocean not sampled by the Argo network).

## 2.2 Data processing

To assess the ocean mass budget, we rely on time series from observations and models. To ensure the consistency between the various datasets, we apply the same processing to each of them. The global means are computed from gridded datasets applying the same restrictive mask so that all budget components cover the same spatial extent. This mask excludes areas not well sampled by Argo data (polar oceans above 60° North and below 60° South and marginal seas) and a buffer zone of 200 km from the coastlines to minimise leakage effects from land mass variations estimated by GRACE and GRACE-FO data (Dobslaw et al., 2020). The mask extent is shown in Figure S1 of the Supplementary Materials. The global averages are weighted according to the surface of sea water within each grid cell. For each time series, annual and semi-annual signals are removed by least-square fitting, a three-month low-pass Lanczos filter is applied and the temporal average is removed. Some components of the budget are assessed from several available estimates. In such cases, an ensemble mean is computed as the average of the considered time series at each time step.

Uncertainties are assumed to be Gaussian and are provided as standard uncertainties, corresponding to the 68 % confidence level. When ensemble means are used to assess a component of the budget, the associated standard uncertainty $\sigma$ is computed by combining the uncertainty $\sigma_{ens}$ from the spread of the datasets included in the ensemble (estimated as the difference between the maximal and the minimal value at each time stamp) and the standard uncertainties $\sigma_{1 \leq i \leq N}$ of the $N$ individual time series when this information is provided, assumed independent:

$$\sigma = \sqrt{\sigma_{ens}^2 + \sum_{i=1}^{N} \sigma_i^2} \tag{3}$$

This approach is used for the contribution of Greenland and Antarctica ice sheets melting and for the thermosteric sea level component. When assessing the budgets, the uncertainties associated with the sum of the components and with the residuals are obtained by summing the variances of the individual components involved.

All linear trends given in this article are computed by an ordinary least-squares regression. The associated uncertainties are estimated using an extended ordinary least-squares method that takes into account the data uncertainties. Trends of the various budget components are provided in Table S1 of the Supplementary Materials. The budget residuals are compared by computing their root mean square errors (RMSEs) provided in Table 1. All RMSEs and trends are computed from 1st January to 31st December of the specified years.

## 3 Data

### 3.1 GRACE/GRACE Follow-On data

We use six GRACE and GRACE-FO solutions from different processing centres, including three mass concentration (mascon) solutions and three spherical harmonics solutions. The mascon solutions are the Release 6 from the Jet Propulsion Laboratory (JPL; Watkins et al., 2015), the Center for Space Research (CSR; Save et al., 2016; Save, 2020) and the Goddard Space Flight Center (GSFC; Loomis et al., 2019). Over the oceans, the mascon solutions are provided as ocean bottom pressure data, with the ocean and atmospheric loading effects included. To obtain the ocean mass change, the spatial mean of the GAD product (Dobslaw et al., 2017), accounting for the static atmospheric surface pressure, is removed from the mascon data (Chen et al., 2019). We also use spherical harmonics solutions up to degree 60, including the Release 6 of the JPL, CSR (Bettadpur, 2018) and German Research Center for Geosciences (GZF; Bettadpur, 2018; Dahle et al., 2018, 2019). These data are provided as Stoke's coefficients of the residual gravitational potential, corresponding to anomalies with respect to modelled atmospheric and ocean effects. The ocean mass change is obtained from the Stokes' coefficients by adding the GAB product to restore the modelled contribution of the dynamic ocean. The Glacial Isostatic Adjustment (GIA) effect is already corrected in the mascon data with the ICE6G-D model (Peltier et al., 2018). We remove the same GIA model from the spherical harmonics solutions. Corrections are also applied for the degree-1 (Swenson et al., 2008; Sun et al., 2016) and degree-2, order-0 (C20) (Loomis et al., 2020) coefficients. No spatial filtering is applied before computing the global mean time series in order to keep the total ocean mass constant.

The individual and ensemble mean GMOM time series based on GRACE and GRACE-FO data are shown in Figure 1. The GMOM time series display a mean trend of $2.19 \pm 0.02$ mm/yr over 2005-2020 as well as important interannual variability mostly related to the El Niño Southern Oscillation (ENSO) events. This is particularly visible during the 2011 La Niña event (Fasullo et al., 2013) that caused a significant negative anomaly, and during the 2015 El Niño event that caused a positive anomaly. All six solutions agree well except in early 2017 where the spherical harmonics solutions appear significantly different from the mascon solutions. This might be due to a higher noise level in the GRACE data (e.g. Fig. 2 from Chen et al., 2022) especially at low degrees (e.g. abrupt changes in the degree-2, order-1 coefficient (C21) over mid-2016 to mid-2017, see for example Fig.7 from Dahle et al., 2019), better removed in mascon solutions using a spatially and temporally variable regularisation (e.g. Loomis et al., 2019). The difference between the mascon and spherical harmonics solutions is shown in Figure S2. No trend difference is observed between the two types of solutions over 2005-2020.

The uncertainty associated with the GRACE and GRACE-FO GMOM time series is computed from the variance-covariance matrix of the ensemble of spherical harmonics solutions constructed by Blazquez et al. (2018) by varying the processing centres and corrections applied for the geocenter motion, Earth oblateness, filtering, leakage and GIA.

### 3.2 Greenland and Antarctica ice sheet data

The amount of ice mass changes from Greenland and Antarctic ice sheets can be estimated from three independent approaches (e.g. Shepherd et al., 2012; Hanna et al., 2013; Cazenave and the WCRP Global Sea Level Budget Group, 2018): (1) volume

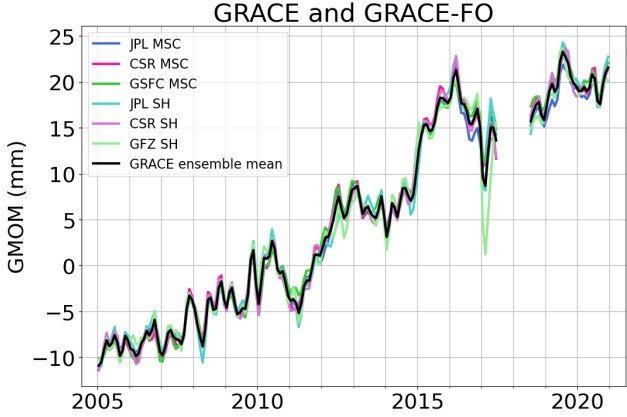

**Figure 1.** Global mean ocean mass time series from GRACE and GRACE-FO mascon (MSC) and spherical harmonics (SH) datasets. The black curve corresponds to the ensemble mean. Linear trends for all time series over different periods of time are provided in Table S1.

change estimated using altimetry data, (2) mass change estimated from gravimetric data, and (3) mass change estimated by the input-output method (IOM) using the surface mass balance from models and the ice discharge measured by InSAR data.

In this study, we consider mostly IOM and altimetry products from different datasets to estimate ice sheet mass loss independently from GRACE and GRACE-FO data. For both Greenland and Antarctica, we use: (1) IOM estimate from Velicogna et al. (2020) and (2) multi-approach estimate from the Ice sheet Mass Balance Inter-comparison Exercise (IMBIE; Shepherd et al., 2021).

IMBIE provides a combination of estimates obtained from the three methods and provides an uncertainty estimate from the spread of the estimates. It is worth noting that the IMBIE product is not independent from GRACE and GRACE-FO data, however, in view of the good agreement between the gravimetric and the altimetric approaches (IMBIE, 2018, 2020; Otosaka, 2021), we choose to include these data in the global mean ocean mass budget assessment. Velicogna et al. (2020) provides estimates from the IOM and compares them with trends adjusted using GRACE data (nevertheless independent from GRACE-FO data). To obtain the sea level contribution from ice mass change, we assume that water is evenly redistributed over the global ocean. Considering a global ocean surface of $361.4 \times 10^8$ km$^2$, 1 Gt of ice is equivalent to 1/361.4 mm of sea level change. In the following, we use ensemble mean time series from the above listed datasets, following the processing described in section 2.2. Figure 2 shows the time series of ocean mass contribution for the individual datasets as well as the ensemble means for Greenland and Antarctica. The ensemble means compared with a purely gravimetry estimate from Velicogna et al. (2020) shows that GRACE and GRACE-FO observations capture a stronger temporal variability (Figure S3).

### 3.3 Land glaciers and ice caps data

To take into account the contribution from glaciers and ice caps to the ocean mass change, we use the recently published data from Hugonnet et al. (2021) covering the 2000-2019 period. The authors used the glacier outlines from the Global Land Ice

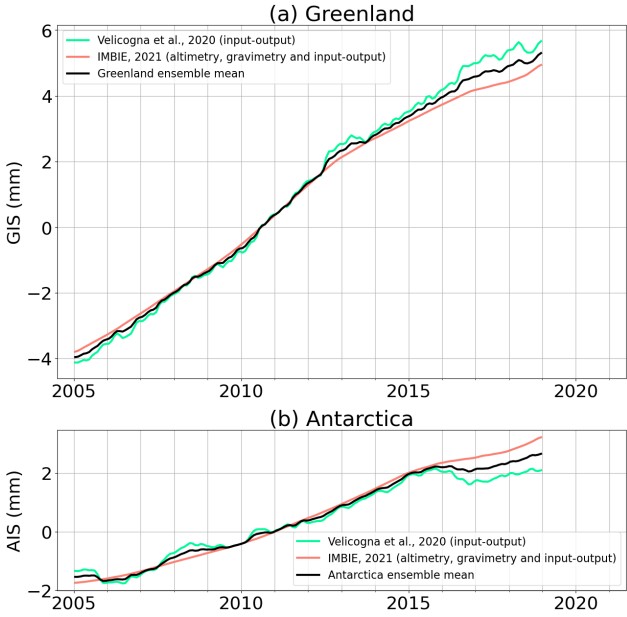

**Figure 2.** Contributions of (a) Greenland ice-sheet (GIS) and (b) Antarctica ice-sheet (AIS) melting to global mean ocean mass change. For each dataset, the method used to estimate the contribution (gravimetry, altimetry or input-output) is indicated in brackets. The black curves correspond to the ensemble means. Linear trends of all time series over different periods of time are provided in Table S1.

Measurements from Space (Tielidze and Wheate, 2018) for the Caucasus Middle East region and from the Randolph Glacier Inventory 6.0 (RGI, 2017) everywhere else. They computed the glacier elevation time series using the following satellite digital elevation models (DEM): ASTER, ArcticDEM and Reference Elevation Model of Antarctica (REMA). The volume change of each glacier was computed with a weighted mean local hypsometric method (McNabb et al., 2019; Hugonnet et al., 2021). For our study, we do not include Greenland glaciers as they are already taken into account in the Greenland ice sheet data (section 3.2).

### 3.4 Terrestrial water storage models

Water stored on land contributes to the changes in global mean ocean mass through the exchange of water between land and oceans. The total terrestrial water storage (TWS) variations result from the water content variations in different reservoirs on land: snow, canopy, soil moisture, groundwater, lakes, reservoirs, wetland and rivers. The change in water content of these reservoirs are driven by both natural climate variability and human activities (e.g. construction of dams on rivers and groundwater abstraction). TWS variations can be estimated from GRACE and GRACE-FO data but here we use global hydrological models independent from gravimetric data.

We consider two hydrological models. The ISBA-CTRIP (Interaction Soil-Biosphere-Atmosphere, Total Runoff Integrating Pathways from the Centre National de Recherches Météorologiques) provides estimates until the end of 2018 (Decharme

et al., 2010, 2019). The WaterGAP (Water Global Assessment and Prognosis) global hydrological model (WGHM) provides data until the end of 2016 (Döll et al., 2003, 2015; Müller Schmied et al., 2020; Müller Schmied et al., 2021). The WGHM provides four estimates of TWS, using two precipitation models and two assumptions on consumptive irrigation water use. The comparison of the ISBA-CTRIP and WGHM models is shown in Figure 3. Over 2005-2016, a trend difference of -0.41 ±

0.24 mm/yr is observed between the two models (Table S1 and Figure 3b). This is likely due to the fact that, unlike WGHM, ISBA-CTRIP does not include the human-induced contributions (HIC) to the TWS estimate. The TWS HIC has become increasingly important over the last decades, reaching a trend of 0.37 (0.30 to 0.45) mm/yr (expressed in sea level equivalent) over 2003-2016 as estimated using WGHM by Cáceres et al. (2020). Adding this trend to ISBA-CTRIP TWS reduces the trend difference between the two models to -0.04 ± 0.28 mm/yr.

In this work, as we aim at understanding the non-closure of the budget after 2016, we use the ISBA-CTRIP model which provides data beyond 2016, and account for the TWC HIC trend estimated by Cáceres et al. (2020). As standard uncertainties, we assign the range of trends provided by Cáceres et al. (2020) over 2003-2016, i.e. 0.13 mm/yr for the climate-driven TWS and 0.15 mm/yr for the human-induced contribution.

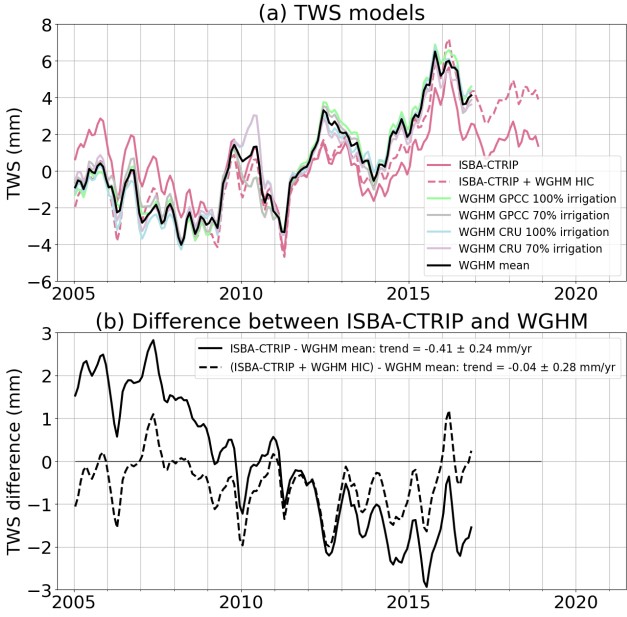

**Figure 3.** Comparison of ISBA-CTRIP and WGHM estimates of terrestrial water storage (TWS) variations. (a) ISBA-CTRIP and WGHM time series. The black curve corresponds to the mean of the four WGHM estimates. The red dotted curve corresponds to the sum of the ISBA-CTRIP climate contribution of the WGHM trend of human-induced contribution (HIC). (b) Difference between ISBA-CTRIP TWS and WGHM mean estimate of TWS. Linear trends of all time series over different periods of time are provided in Table S1.

### 3.5 Atmospheric water vapour

The variations of water content stored in the atmosphere is estimated from the European Centre for Medium-Range Forecasts (ECMWF) atmosphere reanalysis ERA5 (Hersbach et al., 2019) providing the total column water vapour content over both land and oceans. To obtain the sea level equivalent contribution, we assume a uniform distribution of the water volume over the global ocean. Uncertainties are not provided with this component.

### 3.6 Altimetry-based GMSL data

We compute the GMSL time series from the vDT2021 sea level product operationally generated by the Copernicus Climate Change Service (C3S, https://climate.copernicus.eu). This dataset is dedicated to climate-related sea level studies due to the long-term stability of the altimetry missions used to generate the data (Legeais et al., 2021). It provides daily sea level anomalies grids at a 1/4 degree spatial resolution from January 1993 until August 2021, based at any time on a reference altimeter mission (TOPEX/Poseidon, Jason-1, Jason-2, Jason-3 and Sentinel-6 Michaël Freilich), plus a complementary mission (ERS-1,2, Envisat, Cryosat or SARAL/AltiKa depending on the time frame) to increase the spatial coverage. The GMSL time series is corrected for the GIA effect considering a value of -0.3 $\pm$ 0.05 mm/yr (Peltier, 2004) as well as for the sea floor subsidence due to the present-day ice melting with a rate of -0.13 $\pm$ 0.01 mm/yr (Frederikse et al., 2017; Lickley et al., 2018). The GMSL and GMSL trend uncertainties are computed using the uncertainty budget and computational method detailed by Ablain et al. (2019).

Barnoud et al. (2022a) showed that the wet tropospheric correction (WTC) derived from the microwave radiometer (MWR) instrument on-board the Jason-3 satellite, launched in 2016, is likely drifting. This drift was outlined from the comparison of Jason-3's radiometer WTC with a WTC derived from highly stable water vapour climate data records (Schröder et al., 2016) as well as with the radiometer's WTC from the SARAL/AltiKa and Sentinel-3A altimetry missions (Barnoud et al., 2022a, b). The Jason-3 radiometer drift is estimated from the global mean WTC differences between Jason-3, SARAL/AltiKa, Sentinel-3A and the climate data records (Figure 4). The global mean WTC differences show similar low frequency variations. An overall trend of -0.5 mm/yr is observed, but most of the drift is occurring during the first two years of the Jason-3 mission, with a drop of almost 3 mm. This drift results in an overestimation of the GMSL rise since 2016. We compute the average of the three global mean WTC differences that we further use as correction for the GMSL over the Jason-3 period.

### 3.7 Thermosteric sea level data

#### 3.7.1 Argo in-situ data

The global mean thermosteric sea level (GMTSL) is computed from seven in-situ oceanographic datasets: (a) EN4.2.2 data from the Met Office Hadley Center (Good et al., 2013) with Gouretski and Reseghetti (2010) correction applied, (b) IAP (Institute of Atmospheric Physics from the Chinese Academy of Sciences) data (Cheng et al., 2017, 2020), (c) the IFREMER (Institut Français de Recherche pour l'Exploitation de la Mer) ISAS (In Situ Analysis System) 20 dataset (Gaillard et al.,

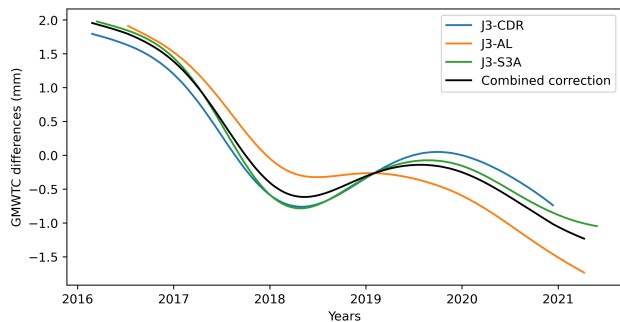

**Figure 4.** Differences between global mean wet tropospheric corrections (GMWTC) from Jason-3 (J3) microwave radiometer (MWR), derived from water vapour climate data records (CDRs), from SARAL/AltiKa (AL) MWR and from Sentinel-3A (S3A) MWR. The average (black curve) of these differences is used as empirical correction for the drift of Jason-3 radiometer.

2016), (d) Ishii et al. (2017) data, (e) JAMSTEC (Japan Agency for Marine-Earth Science and Technology) MOAA GPV (Grid Point Value of the Monthly Objective Analysis using the Argo data) version 2021 product data set (Hosoda et al., 2010), (f) NOAA (National Oceanic and Atmospheric Administration) data (Levitus et al., 2012; Garcia et al., 2019) and (g) SIO (Scripps Institute of Oceanography) data (Roemmich and Gilson, 2009). The seven datasets are mainly based on Argo data (Argo, 2021) for our study period. It is important to note that, over the last few years, delay-mode quality-controlled Argo data are not necessarily available yet, so that mostly real-time data are used in the provided datasets, even though real-time data are not suitable for climate studies. In addition to Argo data, EN4 and IAP datasets also include Mechanical Bathythermograph (MBT) and Expendable Bathythermograph (XBT) data. JAMSTEC includes Triangle Trans-Ocean Buoy Network (TRITON) data and additional conductivity-temperature-depth profiler data from ships.

From these datasets, we compute the thermosteric sea level change due to temperature variations between 0 and 2000 m depth. A linear trend of $0.12 \pm 0.03$ mm/yr is added to the GMTSL to take into account the contribution of the deep ocean (Chang et al., 2019). Figure 5a shows the individual GMTSL time series as well as the ensemble mean.

### 3.7.2 Ocean reanalysis

For comparison with the in-situ thermosteric data, we use the ECMWF ocean reanalysis ORAS5. No available uncertainty are associated with the ORAS5 GMTSL estimate. However, reanalyses have the advantage to use physical modeling in order to provide data over the full ocean, including coastal areas, marginal seas and deep ocean. To enable comparison with the Argo-based GMTSL, the same mask is applied to compute the global mean from ORAS5 data. However, the computation integrates the full water column from 0 to 6000 m, so that we do not need to add the deep ocean linear contribution as for the Argo-based GMTSL. The GMTSL trend from the ORAS5 reanalysis amounts to 1.77 mm/yr which is slightly higher than the one from the Argo in-situ ensemble mean (Figure 5b). Sub-annual and inter-annual GMTSL variations show similar amplitudes between in-situ data and the ORAS5 reanalysis (Figures 5a and b). The ORAS5 data allow us to compute the deep ocean contribution to

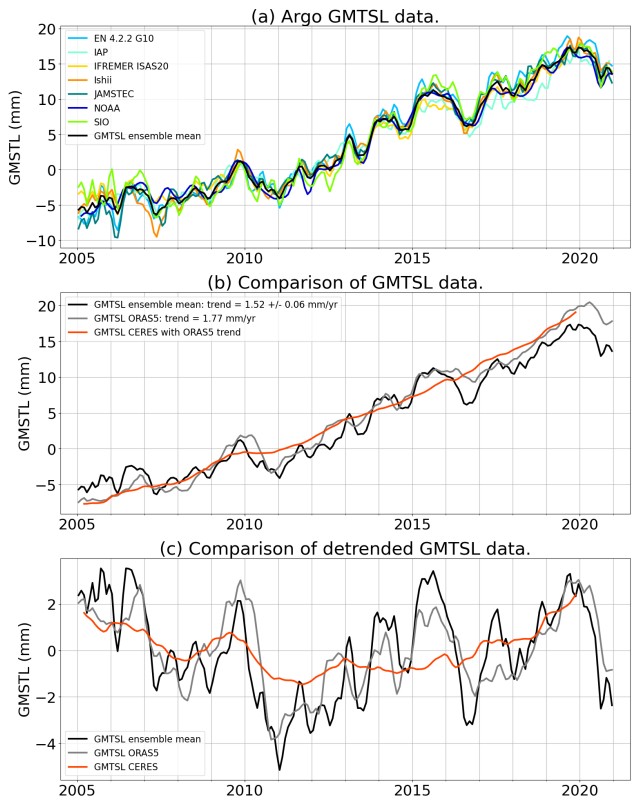

**Figure 5.** Global mean thermosteric sea level (GMTSL) variations time series. (a) GMTSL from the seven datasets based on in-situ measurements used in this study. The black curve corresponds to the ensemble mean. Linear trends of all time series over different periods of time are provided in Table S1. (b) Comparison of the GMTS estimates from in-situ data, ORAS5 reanalysis and CERES observations. (c) Comparison of the detrended GMTSL time series.

the GMTSL (Figure S4): the deep ocean thermosteric contribution is not perfectly linear, but the variations remain negligible compared to the linear estimate from Chang et al. (2019).

### 3.7.3 Top of the atmosphere radiative fluxes measurements

We also use the GMTSL derived from the measurements of the radiative fluxes at the top of the atmosphere by the Clouds and
225 the Earth's Radiant Energy System (CERES) Energy Balanced and Filled (EBAF) instruments (Loeb et al., 2018; Kato et al., 2018). CERES-EBAF data measures the Earth's energy imbalance (EEI). Knowing that about 90 % of the excess of energy is stored as heat into the oceans (von Schuckmann et al., 2020), the ocean heat content change can be derived from the CERES-EBAF EEI, which in turn provides an estimate of the GMTSL (eg. Marti et al., 2022). CERES-EBAF data do not constrain well the EEI mean, hence the GMTSL trend, but provide robust estimate of the interannual variability. In the following, we
constrain the trend of the CERES GMTSL to equal the ORAS5 GMTSL trend. Short-term variations (below 2 to 3 years)

observed by in-situ and reanalysis data are not recovered by CERES measurements (Figures 5c) because CERES only assess the climatic component of thermal variations of the ocean (Palmer and McNeall, 2014). However, the interannual variations are consistent (Figure 5c). No uncertainty is associated with the CERES-based GMTSL estimate. As ORAS5, CERES is used in this study for comparison with the Argo-based GMTSL ensemble mean.

## 4 Resulting global mean ocean mass budgets

### 4.1 Budget from the sum of mass contributions

Figure 6a shows the global mean ocean mass budget comparing the GRACE-based GMOM to the sum of its contributions from ice sheets, ice caps, glaciers, terrestrial water storage and atmosphere water content. The budget residuals are shown in Figure 6b. The drop in the gravimetry-based GMOM in early 2017 is likely linked to the processing of spherical harmonic solutions, as it does not appear when using mascon solutions only as shown in Figures S2 and S5. We computed the RMSE of the residuals over two time spans, 2005-2014 and 2015-2018. These amount to 1.69 mm and 3.15 mm respectively (Table 1). While the 2015-2018 time span is very short, we also estimate a residual linear trend of -1.56 $\pm$ 0.37 mm/yr over these four years (Table S1). The ocean mass budget alone does not allow us to conclude on the stability of the observations. Indeed, the observed residuals could be due either to the gravimetry-based GMOM component or to any of the individual mass contributions estimated by the land ice and water storage variation models. However, recent studies have shown that global hydrological models tend to underestimate interannual and decadal variations in the terrestrial water storage when compared to GRACE and GRACE Follow-On (e.g. Scanlon et al., 2018; Pfeffer et al., 2022b). Thus it is very likely that the larger residuals reported for 2015-2018 compared to the 10-year long time span (2005-2014) reflect interannual uncertainties of the hydrological model used in this study.

**Table 1.** Budget residual root mean square errors (RMSE) in mm. GMSL$_{J3D}$ stands for the altimetry-based GMSL corrected for the Jason-3 WTC drift. GMTSL$_{Argo}$, GMTSL$_{ORAS5}$ and GMTSL$_{CERES}$ stand for the GMTSL estimated from the Argo ensemble mean, from the ORAS5 ocean reanalysis and from CERES observations respectively. RMSE values are computed over common periods of time to enable comparisons (from 1st January to 31st December of the specified years.

| Residual RMSE (mm) | First 10 years 2005-2014 | Last 4-6 years 2015-2018 | Full period 2005-2014 | Figures |
|---|---|---|---|---|
| GMOM-(GIS+AIS+GLA+TWS+AWV) | 1.69 $\pm$ 0.06 | 3.15 $\pm$ 0.55 | 2.21 $\pm$ 0.09 | Fig. 6b |
| GMOM-(GMSL-GMTSL$_{Argo}$) | 1.65 $\pm$ 0.46 | 3.15 $\pm$ 2.85 | 2.18 $\pm$ 0.51 | Fig. S6b |
| GMOM-(GMSL$_{J3D}$-GMTSL$_{Argo}$) | 1.65 $\pm$ 0.46 | 2.46 $\pm$ 2.28 | 1.92 $\pm$ 0.44 | Fig. 7b |
| GMOM-(GMSL$_{J3D}$-GMTSL$_{ORAS5}$) | 2.87 $\pm$ 0.28 | 1.70 $\pm$ 0.59 | 2.59 $\pm$ 0.19 | Fig. 8b |
| GMOM-(GMSL$_{J3D}$-GMTSL$_{CERES}$) | 3.70 $\pm$ 0.37 | 2.14 $\pm$ 0.85 | 3.33 $\pm$ 0.25 | Fig. 9b |
| GMSL$_{J3D}$-(GMTSL$_{Argo}$+GIS+AIS+GLA+TWS+AWV) | 2.05 $\pm$ 0.56 | 5.21 $\pm$ 4.07 | 3.28 $\pm$ 0.71 | Fig. S8b |
| GMSL$_{J3D}$-(GMTSL$_{ORAS5}$+GIS+AIS+GLA+TWS+AWV) | 2.20 $\pm$ 0.17 | 2.58 $\pm$ 0.71 | 2.32 $\pm$ 0.15 | Fig. 10b |

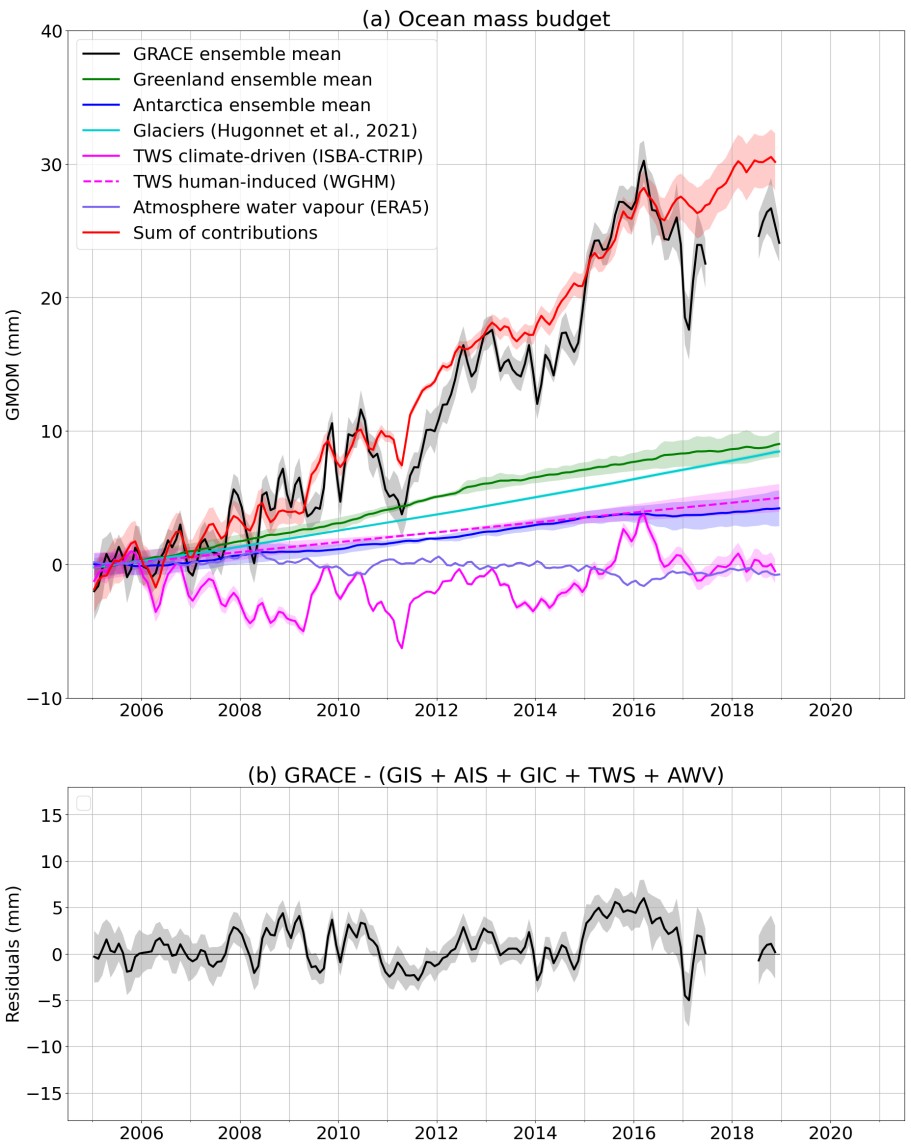

**Figure 6.** Ocean mass budget. (a) Budget with GRACE/GRACE-FO based global mean ocean mass (GMOM) variations and the sum of its contributions from Greenland (GIS), Antarctica (AIS), land glaciers (GIC), terrestrial water storage (TWS) and atmospheric water vapour (AWV) variations. TWS variations are split into the climate-driven component from ISBA-CTRIP and the human-induced contribution trend estimated from WGHM (Cáceres et al., 2020). (b) Budget residuals. Linear trends of all components over different periods of time are provided in Table S1.

## 4.2 Comparing with the altimetry-based GMSL corrected for the thermal expansion

We also compare the gravimetry-based GMOM to the altimetry-based GMSL corrected for the thermal expansion using in-situ oceanographic data. Results are shown in Figure 7 correcting Jason-3 altimetry data for the WTC of Jason-3 MWR drift as of 2016. Figure S6 shows the comparison without the Jason-3 WTC drift correction. Correcting for the Jason-3 WTC drift, the budget residual RMSE over 2015-2018 amounts to 2.46 mm instead of 3.15 mm without correction (Table 1). The budget residual RMSE is reduced by about 22 %. Although this is a significant improvement, the empirical Jason-3 WTC drift correction does not allow closing the mass budget within uncertainties, leaving an unexplained residual drift beyond 2015.

In order to assess the potential impact of the Argo data spatial coverage, of the lack of delay-mode quality-controlled data and of the deep ocean contribution on the estimate of the thermosteric compontent, we reassess the sea level budget using the ORAS5 reanalysis instead of the Argo ensemble mean (Figure 8). Using ORAS5 GMTSL, the budget residual RMSE is reduced to 1.70 mm instead of 2.46 mm with the Argo ensemble mean (Table 1). We note a long period interannual signal in the residuals (Figure 8b) leading to an increase of the RMSEs estimated over the full period (Table 1) even though the residual trend over the full period is not significant (Table S1).

Figure 9 shows the same budget, using CERES GMTSL (with ORAS5 overall trend). The residuals show that CERES GMTSL cannot observe extreme events such as the 2011 La Niña (Figure 9b). In fact, CERES does not observe the subannual variations of GMTSL that are not linked to the long-term storage of heat in the ocean. This lead to higher RMSE values than for ORAS5, but the RMSE over 2015-2018 is improved compared to the budget using Argo data (Table 1). We also observe a long period interannual signal in the residuals, similar to the one observed in the budget using ORAS5 GMTSL.

The sea level budgets using the Argo-based ensemble, the ORAS5 reanalysis and the CERES observations (Figures 7, 8 and 9) suggest that the uncertainties of the GMTSL component are likely to be underestimated, hence that the GMTSL would be responsible for the observed significant residuals. Therefore, the GRACE and GRACE-FO data are unlikely to be responsible on their own for the non-closure of the budget observed in Figures 6 and 7. The non-closure of the sea level budget using the Argo-based GMTSL may be solved in the future when delay-mode quality-controlled data will be fully available and integrated in the gridded datasets.

For completeness, the altimetry-based GMSL corrected for Jason-3 WTC drift is also compared to the sum of the ORAS5 GMTSL contribution and of the individual ocean mass contributions (Figure 10). This is a sea level budget, independent of the GRACE and GRACE-FO GMOM estimate. The residual time series displays some significant interannual variability, especially in 2011 and in 2015-2016 during the ENSO events (Figure 10b). As mentioned in section 4.1, these residual signals may be due to an underestimation of TWS changes at interannual time scales by global hydrological models.

## 5 Conclusions

We compared the GRACE and GRACE-FO based GMOM to the sum of the individual ocean mass contributions and to the GMSL corrected for thermal expansion. Both budgets initially presented a significant residual trend beyond 2015.

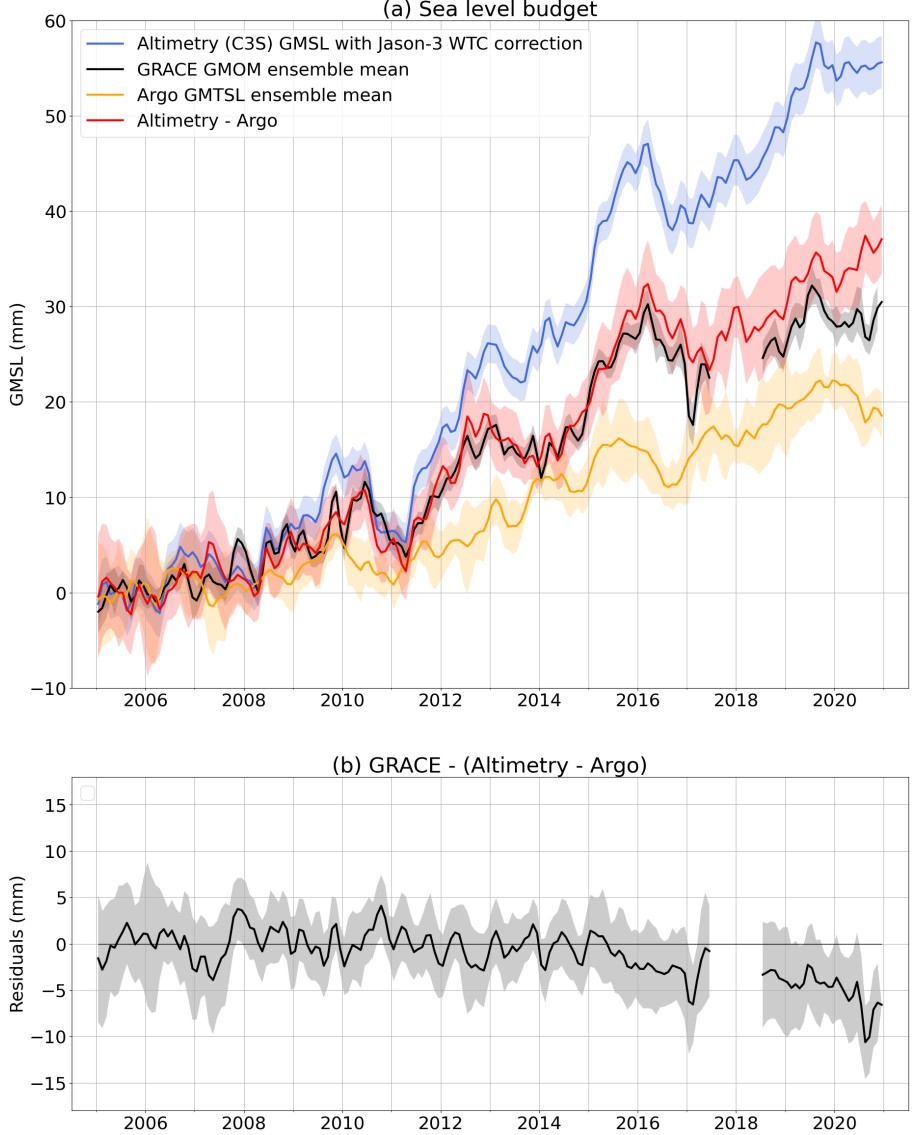

**Figure 7.** Sea level budget. (a) Budget with GRACE/GRACE-FO based global mean ocean mass (GMOM) variations compared to altimetry-based GMSL (corrected for the Jason-3 radiometer WTC drift) and Argo-based GMTSL. (b) Budget residuals. Linear trends of all components over different periods of time are provided in Table S1.

The global mean ocean mass budget comparing the GRACE and GRACE-FO estimate to the sum of individual mass contributions (ice mass changes from ice sheets, ice caps and glaciers, terrestrial water storage variations and atmospheric water vapour variations) shows significant interannual residuals at some periods, in particular during ENSO events (around 2011 and 2015-2016). Such residuals are likely due to an underestimation of TWS changes by global hydrological models at interannual time scales.

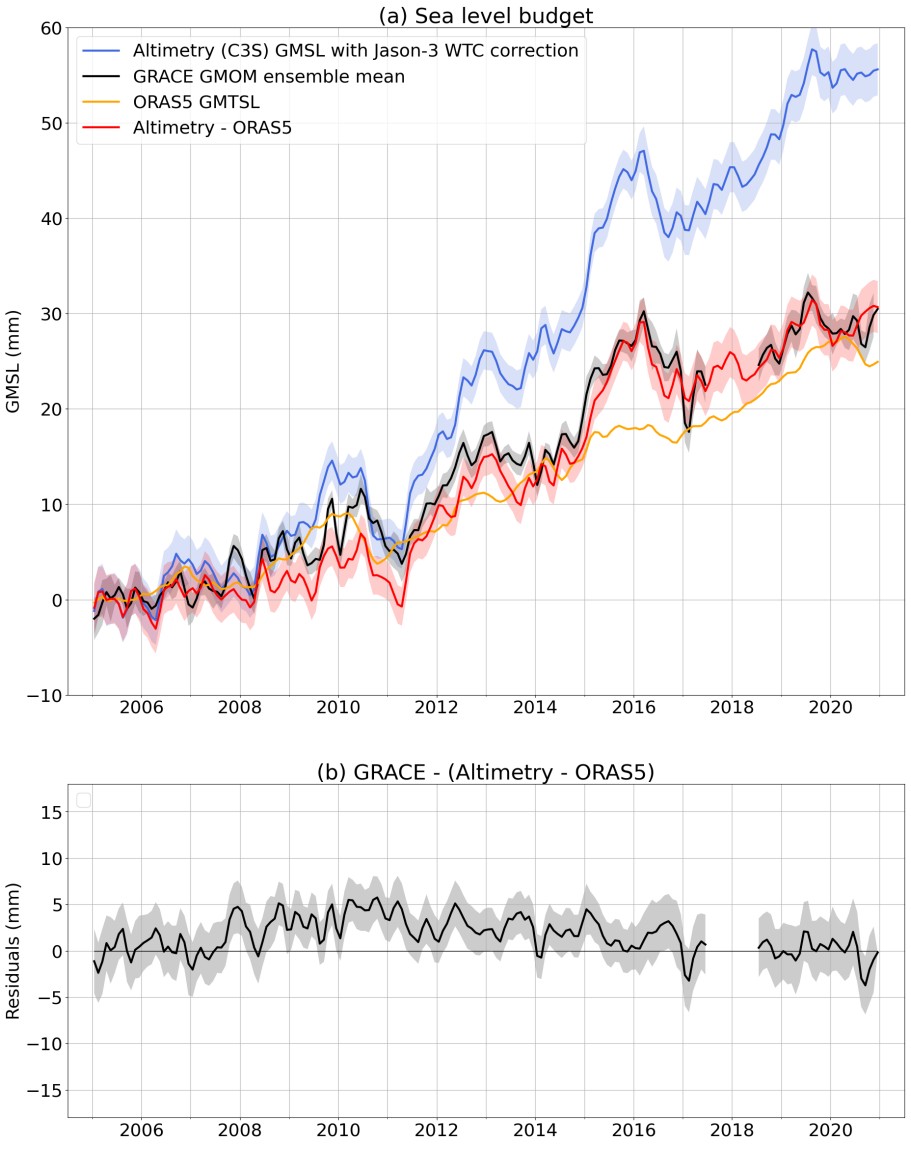

**Figure 8.** Sea level budget. (a) Budget with GRACE/GRACE-FO based global mean ocean mass (GMOM) variations compared to altimetry-based GMSL (corrected for the Jason-3 radiometer WTC drift) and ORAS5 GMTSL. (b) Budget residuals. Linear trends of all components over different periods of time are provided in Table S1.

Comparing the GMOM to the altimetry-based GMSL corrected for the thermal expansion, we showed that a drift of Jason-3 WTC drift is responsible for about 22 % of the budget RMSE over 2015-2018. A correction for Jason-3 WTC drift is estimated based on the WTC from water vapour climate data records as well as from SARAL/AltiKa and Sentinel-3A altimetry missions.
After applying this correction, there are still some significant budget residual remaining.

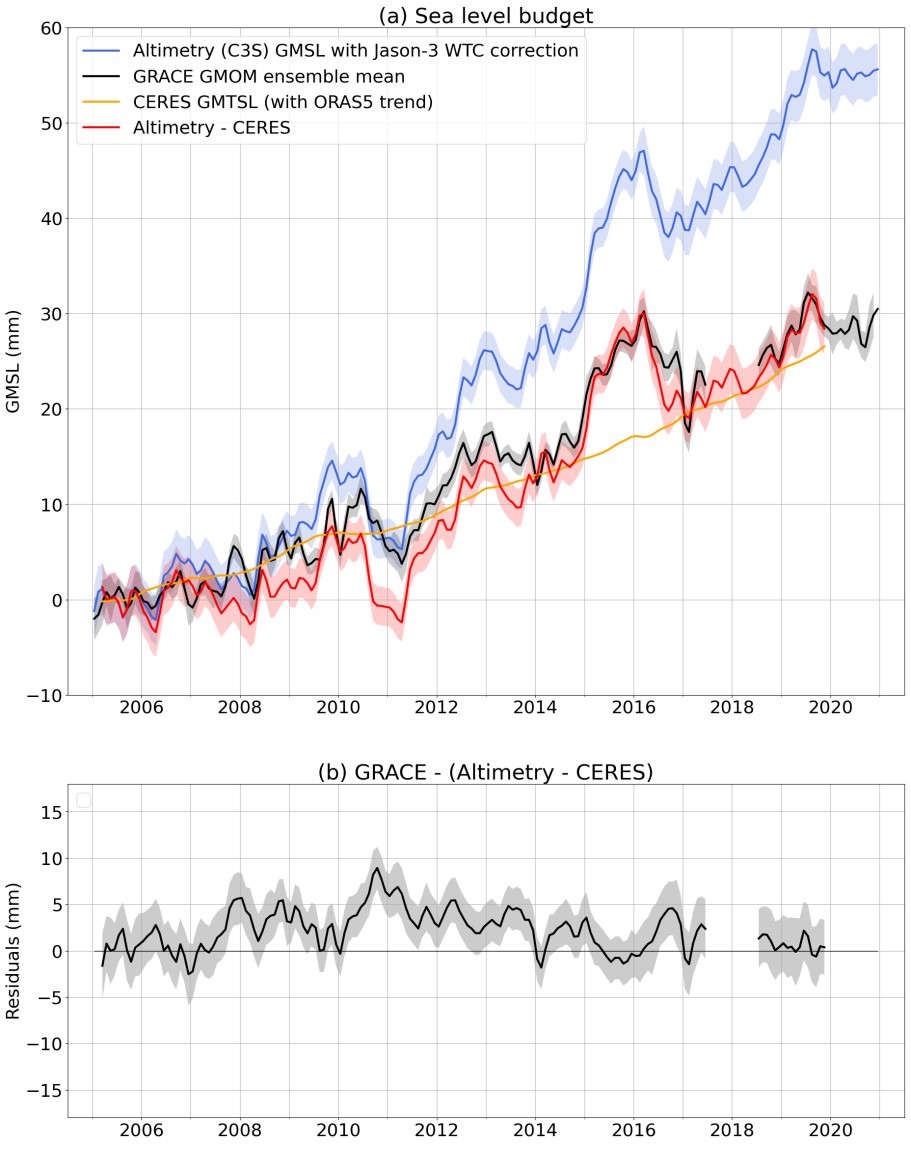

**Figure 9.** Sea level budget. (a) Budget with GRACE/GRACE-FO based global mean ocean mass (GMOM) variations compared to altimetry-based GMSL (corrected for the Jason-3 radiometer WTC drift) and CERES-based GMTSL (using the trend from ORAS5). (b) Budget residuals. Linear trends of all components over different periods of time are provided in Table S1.

Using the thermosteric estimate from the ORAS5 reanalysis and CERES observations instead of Argo data, the sea level budget residuals are significantly reduced in particular over the last years (2015-2020) of our study period. We conclude that recent Argo data are responsible for a major part of the sea level budget residuals, possibly due to the lack of delay-mode quality-controlled data in the gridded products used as input for this study.

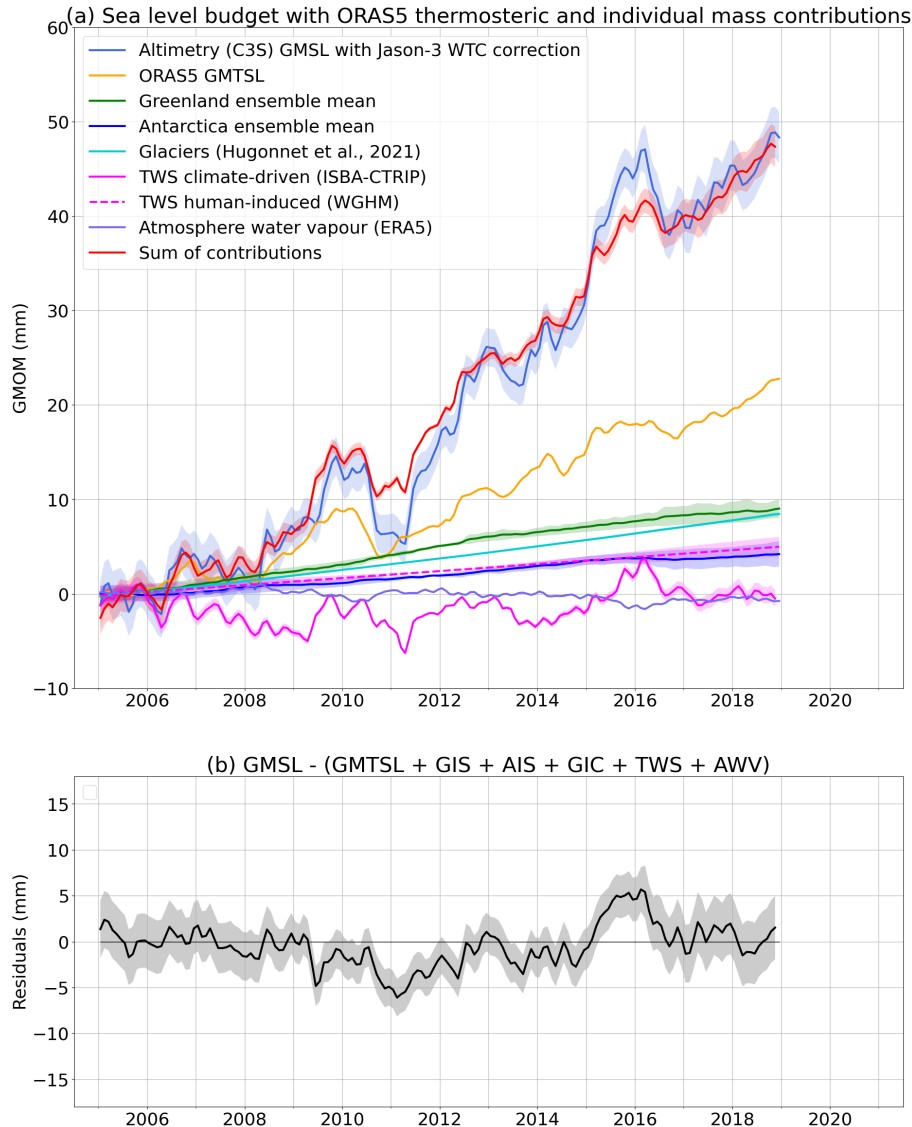

**Figure 10.** Sea level budget using the individual ocean mass contributions. (a) Sea level budget using the ORAS5 GMTSL and taking into account Jason-3 radiometer drift correction. (b) Budget residuals. Linear trends of all components over different periods of time are provided in Table S1.

Finally, the budget comparing altimetry (with the Jason-3 WTC drift corrected) and ORAS5 with the sum of individual mass components still show some interannual signals in the residuals that may be due to a lack of amplitude in the hydrological models during ENSO events.

*Data availability.* The GRACE/GRACE Follow-On JPL, CSR and GFZ Level 2 (Stock's coefficients) and Level 3 (mascon) Release 6 data are available from https://podaac.jpl.nasa.gov. The GRACE/GRACE Follow-On GSFC data are available from https://earth.gsfc.nasa.gov/geo/data/grace-mascons. The C3S altimetry data are available from https://cds.climate.copernicus.eu/cdsapp#!/dataset/satellite-sea-level-global?tab=overview. The AVISO altimetry along-track data for Jason-3, SARAL/AltiKa and Sentinel-3A are available from https://www.aviso.altimetry.fr/. The total column water vapour REMSS climate data records are available at https://wui.cmsaf.eu/safira/action/viewProduktDetails?eid=21864&fid=23. Argo data were collected and made freely available by the international Argo Program and the national programs that contribute to it (https://argo.ucsd.edu, https://www.ocean-ops.org). The Argo Program is part of the Global Ocean Observing System. The NOAA Argo data are available from https://www.ncei.noaa.gov/access/global-ocean-heat-content/. The SIO Argo climatology data are available from http://sio-argo.ucsd.edu/RG_Climatology.html. The JAMSTEC MOAA GPV data are available from http://www.jamstec.go.jp/ARGO/argo_web/argo/?page_id=83&lang=en. The EN4 Argo data are available from http://hadobs.metoffice.com/en4/download.html. The IAP data are available from http://159.226.119.60/cheng/. The IFREMER ISAS20 data are available from https://www.seanoe.org/data/00412/52367/. The Ishii v7.3.1 data are available from https://climate.mri-jma.go.jp/pub/ocean/ts/v7.3.1/. The TWS WGHM v2.2d data are available from https://doi.pangaea.de/10.1594/PANGAEA.918447?format=html#download. The TWS ISBA-CTRIP data are available from the authors (Decharme et al., 2019). For Greenland, Antarctica and glaciers datasets, we invite the reader to refer to the references cited in the article and the links provided therein.

*Author contributions.* AB conducted the study and wrote the initial version of the manuscript. AC supervised the work. JP, MA, RF and VR participated in the data processing and in the discussion of the results. All authors have read, improved and agreed with the content of the manuscript.

*Competing interests.* The authors declare no conflict of interests.

*Acknowledgements.* We thank M. D. Palmer and the second anonymous reviewer for their very interesting comments that helped improve the manuscript and the robustness of the results. The research leading to these results has received funding from the European Research Council (ERC) under the European Union's Horizon 2020 research and innovation program (GRACEFUL Synergy Grant agreement No 855677).

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
