# Peer review of "Revisiting the global mean ocean mass budget over 2005-2020"

_EGUsphere, 2022_

## Author Comment (AC1)

**Overview**

We thank the reviewers for their very interesting comments that helped improve the manuscript and the robustness of the results.

The main changes brought to the manuscript in the revised version are the following:

- We have added the atmosphere water vapour content variations as a component of the GMOM budget using ERA5 data.

- In order to compare with the Argo data, we have reassessed the sea level budget using the ORAS5 reanalysis and CERES data. This allows us to close the budget over the last years, meaning that there is some lack of stability in the Argo data over the last years.

Consequently, GRACE and GRACE-FO data seem unlikely to be responsible for the non-closure observed so far in the sea level and ocean mass budgets. Due to these new comparisons, the discussion and conclusions have significantly changed in the revised version of the manuscript.

Detailed answers to the reviewers comments are provided below in blue (reviewers' comments are in black).

**RC1: 'Comment on egusphere-2022-716', M. D. Palmer, 05 Oct 2022**

This paper deals with closure of the global ocean mass budget for the period 2005-2020 using a combination of observation of model-based data products. This type of study is crucial to our understanding of observed climate change and identifying potential issues / limitations in observing capability and/or data processing. The manuscript is well-written and the figures are of high quality. Dealing with numerous observation and model-based datasets is always challenging in terms of understanding all the potential issues. My main comments below encourage the authors to offer a bit more discussion of the non-budget-closure and to include more quantitative information about how this could be accommodated by the various hypotheses they put forward. They could also consider the relative sizes of estimated uncertainties and/or instances where the uncertainty estimation may be limited by ensemble characteristics, or for other reasons. I find the manuscript to be suitable for publication subject to addressing my comments below.

I'm unsure about the term Global Mean Ocean Mass used throughout the manuscript. I tend to think of this as having units of mass per unit area, but I think we are talking about changes in the total ocean mass? Perhaps there is some explanation or convention that could be mentioned and this point clarified at the start of the manuscript.

=> We use the term "global mean ocean mass change" to designate the global mean sea level change due to the exchanges of freshwater between oceans and continents, including glaciers and ice-sheets. It corresponds to the barystatic sea level (Gregory et al., 2019), but "ocean mass" is also commonly used in the literature to refer to sea level change due to ocean mass change (e.g. Cazenave and the WCRP global sea level budget group, 2018). We have added a sentence at the beginning of the Method section to specify what is called "ocean mass" in the article.

I would like to see the authors spend a bit more discussion on the non-closure of the budgets shown in Figures 6, 7 and 8. Fundamentally, when a budget does not close, it suggests that the uncertainties in one or more components have been underestimated. This point is worthy of some discussion and perhaps some speculation on where limited ensembles or diversity across the

ensemble may be playing a role in the uncertainty estimation. I led a recent paper where we presented a generic framework for using ensembles to characterise uncertainty, which may be of interest to the authors, Palmer et al [2021]:
https://iopscience.iop.org/article/10.1088/1748-9326/abdaec

=> We agree that ensemble spreads provide underestimated uncertainty estimates. Ideally, thorough estimates of uncertainties should be performed. Note that for the altimetry-based GMSL, for the gravimetry-based GMOM and for the glaciers component we use the uncertainties estimated in works dedicated to the uncertainty estimates. Uncertainty estimates of other components are likely to be underestimated, in particular the thermosteric component, but not only. However, this would not explain the change of behaviour over the last year. Non-closure can be due either to underestimated uncertainties or to unidentified systematic errors. In our case, there is some coherent signal in the residuals showing that there is some systematic errors in the last years, not just underestimated random noise.

In the closing sentences of section 4.3, the authors cite "deep ocean below 2000 m depth, the atmospheric water vapour variations and the permafrost thawing" as potential explanations for non-budget closure. I wonder if the authors could offer some more quantitative information in this regard. How large would the temperature variations below 2000 m depth have to account for the residuals, and so on for atmospheric water vapour, permafrost thawing. Would it be helpful if the Argo-based estimates of thermosteric sea-level change could include some estimate of the additional uncertainty below 2000 m? However, I suspect that the horizontal sampling uncertainty may still dominate - and perhaps this is underestimated, as mentioned in my comment on Figure 7. The paper by Allison et al [2019] https://iopscience.iop.org/article/10.1088/1748-9326/ab2b0b neatly illustrates the potential for mesoscale ocean "noise" to introduce spurious signals on a range of timescales, which may be inherent to the observational sampling and common to several (all?) data products? The authors may wish to comment in this regard.

=> We have added the atmospheric water vapour content variations in the GMOM budgets, confirming that this contribution is negligible and does not play a role in the non-closure of the budget. Concerning the deep ocean contribution to the GMTSL, we have added the estimate from the ORAS5 reanalysis in the supplementary materials showing that this contribution seems negligible. In a few years, the data from the Deep Argo array will provide additional information about its role and the variations of temperature below 2000 m. Concerning the permafrost thawing, to our knowledge, no estimates are available and the processes involved are poorly constrained.

The sea level budget now closes over the last years using the ORAS5 data instead of Argo, likely indicating the existence of errors in the Argo-based GMTSL, including errors due to the horiztonal sampling. The coverage of Argo floats have been however analysed and did not indicate any significant change that could explain a biased GMTSL estimate over the last years only (Barnoud et al., 2021, supplementary material). The lack of delay-mode quality-controlled data over the last year is also a possible explanation of the observed change of behaviour between before 2015 and the last years. It may also be noted that ocean reanalyses, such as ORAS5, include a much larger range of measurements, which may help to alleviate specific issues related to Argo.

Figure 6 seems to show a strong correlation between the WGHM  TWS time series and the GRACE ensemble mean. Could these signals have been under- or over- estimated in one of the products? I think this point is worth some discussion. Can the authors comment on the different temporal

resolution of the underlying datasets and ability to resolve the signals? This may also contribute to non-closure. This is mentioned briefly in section 4.3, but discussion of specific timeseries characteristics in section 4.1 and 4.2 could aid the reader.

=> TWS and GMOM are expected to be correlated because TWS variations are the main contribution to subannual time scale GMOM variations and TWS largely contribute to interannual GMOM changes.

Interannual and decadal changes in the terrestrial water cycle may indeed be overlooked in global hydrological models due to inaccurate meteorological forcing (e.g. precipitation), unresolved groundwater processes, anthropogenic influences, changing vegetation cover and limited calibration/validation datasets. Several papers report such issues including Felfani et al. (2017), Scanlon et al. (2018) and Pfeffer et al. (in revision, preprint available online). TWS from global hydrological models is therefore likely to contribute to the non-closure.

- Felfelani, F., Wada, Y., Longuevergne, L., and Pokhrel, Y. N.; Natural and human-induced terrestrial water storage change: A global analysis using hydrological models and GRACE, Journal of Hydrology, 553, 105-118, 2017.

- Scanlon, B. R., Zhang, Z., Save, H., Sun, A. Y., Müller Schmied, H., Van Beek, L. P., ... and Bierkens, M. F.; Global models underestimate large decadal declining and rising water storage trends relative to GRACE satellite data, Proceedings of the National Academy of Sciences, 115(6), E1080-E1089, 2018.

- Pfeffer, J., Cazenave, A., Blazquez, A., Decharme, B., Munier, S., and Barnoud, A.: Detection of slow changes in terrestrial water storage with GRACE and GRACE-FO satellite gravity missions, EGUsphere [preprint], https://doi.org/10.5194/egusphere-2022-1032, 2022

Figure 7(a): The uncertainty on the in-situ thermosteric ensemble mean is much smaller than I would have expected. Was this informed simply from ensemble spread, as shown in Figure 5? Palmer et al [2021] argues that "structural uncertainty" from ensemble spread needs to be combined with some estimate of "internal/parametric uncertainty" in order to fully characterise the total uncertainty. I would encourage the authors to give this standpoint some consideration and update the uncertainty estimate if appropriate.

=> Yes, the uncertainty estimate for the thermosteric component only comes from the ensemble and is therefore likely to be underestimated. We thank the reviewer for the very interesting work of Palmer et al. (2021). To limit the underestimation and given the available information we have from the input data, we now use the difference between the maximum and minimum estimates at each time stamp rather than the standard deviation as the ensemble is constituted of only seven time series.

Figure 8: I don't see panels (c) and (d) in this figure, as implied by the figure caption. I think perhaps the caption descriptions for (c) and (d) are intended to apply to panels (a) and (b)? Similar to Figure 6, there are some apparent correlations between the residuals and the TWS timeseries in particular. One thing to be cautious of is that fact that delayed-mode quality control of Argo floats typically takes 1-2 years to complete. Therefore, the last 1-2 years of data can be considered "provisional" and may be subject to revision, although I think this is generally considered more of

an issue for salinity data, as noted here https://floats.pmel.noaa.gov/float-data-delayed-mode-quality-control

=> We have corrected the caption of Figure 8 (now Figure 10). Yes, the thermosteric component is a potential candidate for sources of uncertainties or errors over the last years due to the lack of delay-mode quality-controlled (DMQC) data. Besides, it may take even longer than 1-2 years for the data providers to replace the real time data by the DMQC data in the gridded products.

We have looked at the correlations between the budget residuals and each component. It is difficult to draw a conclusion from these comparisons because it appears that most components (GMSL, GMOM, GMTSL and TWS) shows correlations with the budget residuals at high frequencies / interannual variations (corresponding to ENSO events in particular), see, for instance, 2009-2011, 2015-2016).

Figure 9: I'm not sure I understand the plot titles.The summation symbol would tend to suggest to me that the quantity subtracted is always (GIS + AIS + GIC + TWS), but this is not this is not the case for panel (c)?  I think the similarity in the timeseries shown in Figure 9(b) strongly implies that GRACE and (GIS + AIS + GIC + TWS) must have similar timeseries, as shown in Figure 9(c), so I'm not sure how much additional information this really offers the reader. In addition to the trends, are there physical insights we can draw on from the variations/similarities in the residuals? In the figure caption, please clarify the precise period that trends are calculated over - e.g. 1st Jan 2015 to 31 Dec 2018 or similar.

=> We have removed this figure for clarity. The summation symbol actually stood for other components, depending on the budget considered, and specified in the plot legends. In the previous version of the manuscript, the idea of this Figure was to show that there were significant residuals in all three budget configurations, showing that there were some errors left in at least two (possibly more) components.

We have specified in the method section that trends are computed from 1$^{st}$ January to 31$^{st}$ December.

Line 5: On the residual trend, it's helpful to be explicit on whether the GRACE-determined mass trend is larger or smaller than the sum of individual components.

=> We have added that the GRACE-based estimate is lower than the estimate from the sum of components.

Line 12: Stylistic choice, but I would recommend replacing "Besides" with "In addition". Same sentence, suggest replacing "water vapour" with "water content".

=> OK, modified.

Line 15: Please cite the latest IPCC AR6 report and specify a period for which the two-thirds statement applies (this has changed over time), as noted in the Working Group I summary for policymakers and Chapter 9 (Fox-Kemper et al, 2021).

=> We have added the reference and specified the period (2006-2018).

Line 30: Typo? Replace "float" with "floats".

Line 56: Replace "by  the Argo float" with "by Argo floats".

=> We have replaced "Argo float" by "Argo network".

Line 90: Could you briefly comment on the choice of GIA dataset and what effect a different dataset might have on your analysis? Some idea of the importance of this for the reader would be helpful.

=> The effect of the choice of the GIA is included in the uncertainty estimate following Blazquez et al., 2018 (see end of paragraph 3.1). Note that it would only affect the linear trend and not the high frequencies or interannual signals.

Lines 96-97: Can you comment on the physical plausibility of some of the very sharp drops in the datasets seen in 2017? E.g. what would this imply for rainfall over land and subsequent river flows? How do these timeseries compare with timeseries of terrestrial land water storage shown later in the manuscript? I suspect that this cursory analysis would support "noise" as the main candidate explanation.

=> The drop is much more limited in the mascon solution than in the SH solutions, and it is not observed at all in the TWS at this time. The most likely reason is indeed the level of noise of the gravimetry-based GMOM, not only because of the noise level of the measurements themselves linked to the low rate of available data in the second half of 2016 and the loss of one accelerometer from the end of 2016. Even during some extreme ENSO events, the amplitudes variations are not that high. We cannot exclude that this drop comes from some physical processes, even though it is not observed in the hydrological model TWS.

Equation (1) and (2): I would suggest a different notation for Epsilon between these equations, perhaps Epsilon1 and Epsilon2. This would make clear that these quantities are fundamentally different (a dash is often used when one quantity is a proxy for the other) - they are related to completely independent datasets (?)

=> We have replaced epsilon and epsilon' by epsilon1 and epsilon2 as suggested.

Equation (3): I don't understand why the standard uncertainties are raised to the power 3 before summing them. Is there a reference you can cite that explains this approach? Or offer some additional explanation.

=> This was a typo (it should have been squared) and it has been corrected.

Figure 1: Please include an explanation of the units of mass in global mean sea level equivalent. A second y-axis in units of Gt or similar could usefully be included.

Figure 2 : Same comment as for Figure 1 applies here. Please consider this point for all subsequent figures (may be more appropriate to some figures than others).

=> We now define what we call "ocean mass" in the article at the beginning of section 2.1. The conversion between Gt of ice and sea level equivalent used for the Greenland and Antarctica contributions is detailed in section 3.2.

**RC2: 'Comment on egusphere-2022-716', Anonymous Referee #2, 17 Oct 2022**

I have read the paper by Barmoud dealing with closure of the global ocean mass budget for the period 2005-2020. I agree with the other author that such study is crucial to our understanding of observed climate change and identifying potential problems and limittions in observing capability and/or data processing. In general, the manuscript is well-written, but there are a number of issues

and clarifications that need to be dealt with. I also agree with the other reviewer to enc encourage more discussion of the non-budget-closure with respect to the various hypotheses presented in the paper.

One initial concern deals with the datasets. They have various coverage, and as such this needs to be accounted for in the comparison. One example is the altimetric ocean dataset. Apparently, this dataset is limited to 60N where as previous investigations have been limited to 66N. What is the reason for this limitation? It can not be the 200 km distance to shoreline but some other argument?

If the investigation is limited to 66S-6ON then all major contributors to mass changes are outside the ocean mask. Hence the authors NEED to revise the manuscript and compute the sea level fingerprints as all of the contributing datasets (GIS and AIS in particular) is completely outside this limitation. In my view, this is a requirement to perform this before the manuscript is published, as it might have a significant impact on the results.

=> We thank the reviewer for noticing this limiting mask. Indeed, by limiting the study area to the areas covered by Argo data, the latitudes are limited to +60° North. We have corrected the description of the mask in the manuscript. Note that not taking into account the fingerprints cannot explain the misclosure of the budget occurring from 2015, but would impact the time series over the full period in the same way.

We have compared the impact of fingerprints on the estimate of GMOM using various latitudinal masks (cf. figure below). We can see that the latitudinal mask has a negligible influence on the GMOM estimate, with differences of at most 0.04 mm/yr between a +/-60° mask and no mask at all, i.e. including all latitudes up to 90°.

[Figure]

Figure 1: Comparison of GMOM with different masks (+/- 60°, +/-66° and +/-90° i.e. no mask) applied to the absolute sea level fingerprints computed by Adhikari et al. (2019) using JPL, CSR and GFZ data.

Figure 6. I agree with the other reviewer that there is something wrong with the residuals. I also noticed that the computations/comparisons in this figure, unfortunately, ends sometimes in 2018 which really calls for an update to the time series before publication.

=> Unfortunately, it is not possible to update this budget beyond 2018. We depend on the availability of published datasets for each component. In particular, we are here limited by the availability of the hydrological models independent from gravimetric data.

Equation (3): I agree with the other reviewer that there is something wrong with this equation.

=> Thank you for spotting the error. There was a typo and it has been corrected.

Section 3.3. The paper claims that Glaciers in Greenland is left out because they are already a part of the Greenlandic estimates. This is, to my knowledge, incorrect. At least they are not a part of the estimates in Simonsen 2021 and Mankoff, 2021). This needs further clarification.

=> We thank the reviewer for spotting this inconsistency. In the revised version of the article, we have removed the estimates from Simonsen et al. (2021) and from Mankoff et al. (2021). This has hardly no impact on the GIS contribution ensemble mean estimate.

The authors devote large parts to the discussion on the contribution of a possible trend in the Jason-3 MWR/WTC being responsible of up to 40% of the differences. This is a critical point in the paper as it is referred to unpublished material by Bernaud, 2022 as a lot of the following discussion is related to Jason-3 issue.

=> Another article on the wet troposphere correction derived from water vapour climate data records should be published soon (Barnoud et al., minor revisions, to be resubmitted soon). We have added another reference to some online material (Barnoud et al., 2022, OSTST science team meeting, 10.24400/527896/a03-2022.3403).

A closer look at their own figure 6 brings me seriously doubt about this explanation that Jason is really the problem. Particularly the lower part of Figure 7 indicates that the difference between altimetry and other mass-contribution clearly diverged from late 2014/early 2015. This is more than a year before the launch of Jason-3 in 2016 delivering reliable data from March/April 2016 Wouldn't this means that a more intuitive explanation would be that the older Jason-2 started drifting during its old age and the problem being that the tandem mission correction of the MWR between Jason-2->3 was in error?.

=> There is no contradiction here. The Jason-3 WTC drift was not identified from budget analysis but from comparisons with independent WTC from SARAL/AltiKa and Sentinel-3A as well as from a robust WTC estimated from water vapour climate data records. We do argue that it is an identified problem that can and should be corrected for. However, we do not claim that Jason-3 drift is the only problem occurring and we do not claim that it occurs over the full period of non-closure of the budget.

A potential drift of Jason-2 is perfectly possible. However, it has not been possible to show such a thing yet and it would be purely speculative at this stage while we have been able to detect the Jason-3 drift and estimate an empirical correction for it thanks to the available datasets.

In my view the authors explanation of a drift in the Jason-3 radiometer is very vague. Particularly as the authros discuss the significant trend in the 2015-2018 period. During this period Jason-3 was only present 65% of the time (2016-04-2018.12) . If Jason-3 is responsible for 40% of the trend in this period the apparent trend in Jason-3 during its presence (in 65% of the time series) much consequently have been much larger. This should also be addressed in more detail.

=> The residual trends are computed over fixed periods of time (either 2015-2018 or 2015-2020) to enable comparisons as much as possible. Indeed, the Jason-3 WTC drift only affects data from April 2016.

Similarly to reviewer 1 I have an issue with the physical plausibility of the very sharp drops in the datasets seen in 2017? Please explain this. Could this be related to the missing GRACE-GRACE_FO during this period?.

=> Yes, it is most probably due to the low rate of available data and the high noise level due to the loss of one of the two accelerometers at the end of GRACE mission. It is unlikely not to be physical (even though it cannot be excluded) as shown by the difference between mascon and spherical harmonics solutions. This is why we show the GRACE-based GMOM and the budgets for the two different kinds of solutions in supplementaries. This is discussed in paragraph 3.1.

When it comes to the discussion points in line 238-242 that potential evolution below 2000 m depth, permafrost thawing, and atmospheric water vapor, but In line 190 the authors already investigated and corrected for the deep ocean contribution which ranges up of 0.1 mm/year. Again this magnitude is very small compared to the difference seen, so I do not follow this argumentation.

=> We agree that all these contributions are expected to be very small. We have now included the estimates of the deep ocean thermosteric contributions (with the ORAS5 reanalysis) and the atmospheric water vapour contribution (using ERA5). The discussion has changed as we now show, using the ORAS5 thermosteric estimate, that the non-closure of the sea level budget is likely to be due to Argo thermosteric estimates over the last years.

All in all I find the issue on revising the global ocean mass budget extremely important but the paper and findings are presently not adequately convincing for publication.

Without computing the full fingerprints of the contribution to deal with the limited ocean mask I do not think that the paper presents substantial clear and new information. Particularly as many results are only presented up to 2018.

I suggest the authors to revisit the data perform the correct computation and extend the timeseries as much as possible so the paper and the conclusions could really represent the 2005-2020 period.

=> As shown above, the fingerprints play a negligible role in the computation of the independent mass contributions to sea level change.

As explained above, we are limited by the availability of the hydrological model independent from gravimetric data so that we cannot update the global mean ocean mass budget (Figure 6) beyond 2018 for now. If we could have extended the time series, we would have done so. All trends are provided over the common period of availability (2005-2018) to enable comparisons and we have chosen to provide trends up to 2020 as well for the comparison with altimetry and Argo data as datasets were available.

The revised version of the manuscript presents major advances in our understanding of the non-closure of the budget over the recent years, including the quantification of the Jason-3 wet troposphere correction drift and the role of the Argo thermosteric sea level estimates. We hope that our answers to the reviewers' comments and the corrections made to the manuscript have clarified the article and addressed the reviewer's concerns.